# Improving Diffusion-based Data Augmentation with Inversion Circle Interpolation

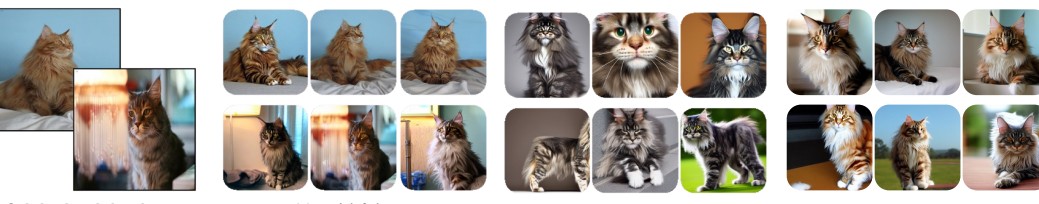

| Original training images | (a) Faithfulness | (b) Diversity | (c) Faithfulness and diversity |

Figure 1: Based on given training images, data augmentation (DA) aims to generate new **faithful** and **diverse** synthetic images. (a) These synthetic images are faithful but not diverse. (b) These synthetic images are diverse but not faithful. (c) These synthetic images are both faithful and diverse.

## Abstract

Data Augmentation (DA), *i.e.*, synthesizing faithful and diverse samples to expand the original training set, is a prevalent and effective strategy to improve various visual recognition tasks. With the powerful image generation ability, diffusion-based DA has shown strong performance gains on different benchmarks. In this paper, we analyze today's diffusion-based DA methods, and argue that they cannot take account of both *faithfulness* and *diversity*, which are two critical keys for generating high-quality samples and boosting final classification performance. To this end, we propose a novel Diffusion-based Inversion Interpolation DA method: **Diff-II**. Specifically, Diff-II consists of three main steps: 1) *Category concepts learning*: Learning concept embeddings for each category. 2) *Inversion interpolation*: Calculating the inversion for each image, and conducting random circle interpolation for two randomly sampled inversions from the same category. 3) *Two-stage denoising*: Using different prompts to generate synthesized images in a coarse-to-fine manner. Extensive experiments on multiple image classification tasks (*e.g.*, few-shot, long-tailed, and out-of-distribution classification) have demonstrated its effectiveness over state-of-the-art diffusion-based DA methods.

## 1 Introduction

Today's visual recognition models can even outperform us humans with sufficient training samples. However, in many different real-world scenarios, it is not easy to collect adequate training data for some categories. For example, since the occurrence frequency of various categories in nature follows a long-tailed distribution, there are many rare categories with only limited samples (O'Hagan & Forster, 2004; Van Horn & Perona, 2017). To mitigate this data scarcity issue, a prevalent and effective solution is *Data Augmentation (DA)*. Based on an original training set with limited samples, DA aims to generate more synthetic samples to expand the training set.

For DA methods, there are two critical indexes: *faithfulness* and *diversity* (Sajjadi et al., 2018). They can not only evaluate the quality of synthesized samples, but also influence the final recognition performance. As shown in Figure 1, **faithfulness** indicates that the synthetic samples need to retain the characteristics of the corresponding category (*c.f.*, Figure 1 (a)), *i.e.*, the faithfulness confirms that the model learns from correct category knowledge. **Diversity** indicates that the synthetic samples should have different contexts from the original training set and each other (*c.f.*, Figure 1 (b)), *i.e.*, the diversity ensures that the model learns the invariable characteristics of the category by seeing diverse samples.

Figure 2: a) **Intra-category DA**: Given a reference image (from the original set), it adds some noise and denoises with a prompt containing the same category concept (*e.g.*, concept "`[A]`" for category A image). (b) **Inter-category DA**: Different from Intra-category DA, it denoises with a prompt containing a different category concept (*e.g.*, concept "`[B]`" for category A image). (c) **Ours**: It first calculates the inversion for each image, and conducts random circle interpolation for two images of the same category. Then, it denoises in a two-stage manner with different prompts.

With the photo-realistic image generation ability of today's diffusion models (Ho et al., 2020; Nichol & Dhariwal, 2021), a surge of diffusion-based DA methods has dominated the image classification task. Typically, diffusion-based DA methods reformulate image augmentation as an image editing task, which consists of two steps: 1) *Noising Step*: They first randomly sample an image from the original training set as a reference image and then add some noise to the reference image. 2) *Denoising Step*: They then gradually denoise this noisy reference image conditioned on a category-specific prompt. After the two steps, a new synthesized training image was generated. Following this framework, the pioneer diffusion-based DA work (He et al., 2022) directly uses a hand-crafted template containing the reference image's category label as the prompt (*i.e.*, intra-category denoising). These handcrafted prompts work well on general datasets with a broad spectrum of category concepts (*e.g.*, CIFAR-10 (Krizhevsky et al., 2009)). However, these few words (with only category name) can not guide the diffusion models to generate images with specific and detailed characteristics, especially for datasets with fine-grained categories (*e.g.*, Stanford Cars (Krause et al., 2013)).

To further enhance the generalization ability, subsequent diffusion-based DA methods try to improve the quality of synthesized samples from the two key characteristics. Specifically, **to improve faithfulness**, Trabucco et al. (2023) replace category labels with more fine-grained learned category concepts. As shown in Figure 2(a), they first learn a specific embedding "`[A]`" for "category A" bird, and then replace the fixed category name with the learned concept in the prompt. These learnable prompts can somewhat preserve fine-grained details for different categories. However, the fixed combination of a reference image and its corresponding category concept always results in similar synthetic samples (*i.e.*, limited diversity). On the other side, **to improve diversity**, Wang et al. (2024) use prompts containing different category concepts (*e.g.*, "`[B]`") from the reference image (*i.e.*, inter-category denoising). This operation can generate images with "intermediate" semantics between two different categories. However, it inherently introduces another challenging problem to obtain an "accurate" soft label for each synthetic image, which affects faithfulness to some extent (*c.f.*, Figure 2(b)). Based on these above discussions, we can observe that: *current state-of-the-art diffusion-based DA methods cannot take account of both faithfulness and diversity*, which results in limited improvements on the generalization ability of downstream classifiers.

In this paper, we propose a simple yet effective **Diff**usion-based **I**nversion **I**nterpolation method: **Diff-II**, which can generate both faithful and diverse augmented images. As shown in Figure 2(c), Diff-II consists of three steps: 1) *Category Concepts Learning*: To generate faithful images, we learn a specific embedding for each category (*e.g.*, "`[A]`" for category A) by reconstructing the images of the original training set. 2) *Inversion Interpolation*: To improve diversity while maintaining faithfulness, we calculate the inversion[1] for each image of the original training set. Then, we sample two inversions from the same category and conduct interpolation. The interpolation result corresponds to a subsequent high-quality synthetic image. 3) *Two-stage Denoising*: To further improve

---

[1]In the image generation field, the inversion refers to a latent representation that can be used to reconstruct the corresponding original image by the generative model.

the diversity, we prepare some suffixes[2] (*e.g.*, "*flying over water*", "*standing on a tree branch*") that can summarize the high-frequency context patterns of the original training set. Then, we split the denoising process into two stages by timesteps. In the first stage, we denoise the interpolation results guided by a prompt containing the learned category concept and a randomly sampled suffix, *e.g.*, "a photo of a [A] bird [suffix]." This design can inject perturbation into the early-timestep generation of context and finally contributes to diversity. In the second stage, we replace the prompt with "a photo of a [A] bird" to refine the character details of the category concept.

To be specific, we first utilize some parameter-efficient fine-tuning methods (*e.g.*, low-rank adaptation (Hu et al., 2021) and textual inversion (Gal et al., 2022)) to learn the concept embedding for each category. Then, we acquire the DDIM inversion (Song et al., 2020) for each image from the original set conditioned on the learned concept. After that, we randomly sample two inversions within one category as one pair and conduct interpolation with random strengths. To align the distribution of interpolation results with standard normal distribution and get a larger interpolation space, we conduct random circle interpolation. Since each pair of images used for inversion interpolation belongs to the same concept, their interpolations will highly maintain the semantic consistency of this concept (*i.e.*, it ensures faithfulness). Meanwhile, since both images have different contexts, the interpolations will produce an image with a new context (*i.e.*, it ensures diversity). Finally, we set a *split ratio* to divide the whole denoising timesteps into two stages. In the first stage, we use a prompt containing the learned concept and a randomly sampled suffix[2] to generate noisy images with diverse contexts (*e.g.*, layout and gesture). In the second stage, we remove the suffix to refine the character details of the category concept. By adjusting the *split ratio*, we can control the trade-off between faithfulness and diversity. To extract all suffixes, we first utilize a pretrained vision-language model to extract all captions of the original training set, and then leverage a large language model to summarize them into a few suffixes.

We evaluated our method on various image classification tasks across multiple datasets and settings. Extensive results has demonstrated consistent improvements and significant gains over state-of-the-art methods. Conclusively, our contributions are summarized as follows:

- We use a unified view to analyze existing diffusion-based DA methods, we argue that they can not take account of both faithfulness and diversity well, which leads to limited improvements.
- We propose Diff-II, a simple yet effective diffusion-based DA method, that leverages the inversion circle interpolation and two-stage denoising to generate faithful and diverse images.
- We conduct comprehensive experiments on three tasks. Our state-of-the-art performance verifies that Diff-II can achieve effective data augmentation by generating high-quality samples.

## 2  RELATED WORK

**Diffusion-Based DA.** With the emergence of diffusion models, Diffusion-based DA (Michaeli & Fried, 2024; Islam et al., 2024) is increasing. Currently, there are two main paradigms of Diffusion-based DA: 1) *Latent Perturbation* (Zhou et al., 2023; Fu et al., 2024; Zhang et al., 2024) generate samples by perturbating latent codes in the latent space. Although these methods can generate diverse samples, due to the uncontrollable perturbation direction, the generated results sometimes deviate from the domain of the original dataset. Therefore, they heavily rely on extra over-sampling and filtering steps. 2) *Image Editing* (He et al., 2022; Trabucco et al., 2023; Dunlap et al., 2023; Wang et al., 2024) reformulate data augmentation as an image editing task (Meng et al., 2021; Lu et al., 2023). However, due to the limitations of the editing paradigm, it is difficult for them to take into account both the faithfulness and diversity of the synthetic samples. To tackle the above problem, our work proposes to generate new images by interpolating the inversions.

**Interpolation-Based Data Augmentation.** For time series and text data, interpolation is a common approach for DA. Chen et al. (2022) incorporate a two-stage interpolation in the hidden space to improve the text classification models. Oh et al. (2020) propose to augment time-series data by interpolation on original data. In the computer vision community, there are some studies (DeVries & Taylor, 2017; Zhou et al., 2023) work on interpolation-based DA for image classification. However, how to combine the excellent generation ability of diffusion models and interpolation operation to obtain high-quality synthetic samples remains an important challenge. To solve this problem, we propose Diff-II, an efficient diffusion-based DA framework with inversion interpolation.

---

[2]More details are left in the appendix.

Figure 3: **Pipeline of Diff-II**. (1) Concept Learning: Learning accurate concepts for each category. (2) Inversion Interpolation: Calculating DDIM inversion for each image conditioned on the learned concept. Then, randomly sampling a pair and conducting random circle interpolation. (3) Two-stage Denoising: Denosing the interpolation results in a two-stage manner with different prompts.

## 3 METHOD

**Problem Formulation.** For a general image classification task, typically there is a **original training set** with $K$ categories: $\mathcal{O} = \{\mathcal{O}^1, \mathcal{O}^2, ..., \mathcal{O}^K\}$, where $\mathcal{O}^i$ is the subset of all training samples belong to $i_{th}$ category. For $\mathcal{O}^i$, there are $N_i$ labeled training samples $\{X_j^i\}_{j=1}^{N_i}$. The classification task aims to train a classifier with $\mathcal{O}$ and evaluate it on the test set. On this basis, diffusion-based DA method first generates extra synthetic images for each category. The **Synthetic set**: $\mathcal{S} = \{\mathcal{S}^1, \mathcal{S}^2, ..., \mathcal{S}^K\}$, $\mathcal{S}^i$ is the set of synthetic images of $i_{th}$ category. Then it trains an improved classifier with both original and synthetic images (*i.e.*, $\mathcal{O} \cup \mathcal{S}$).

**General Framework.** As shown in Figure 3, our proposed Diff-II consists of three main steps:

1) Category Concepts Learning (Sec. 3.1): We first set $n$ learnable token embeddings for each category, and insert some learnable low-rank matrixes into the pretrained diffusion U-Net. By reconstructing the noised image of the original training set $\mathcal{O}$, we learn the accurate concept for each category. We denote the tokens of the $i_{th}$ category concept as $\{[V_j^i]\}_{j=1}^n$.

2) Inversion Interpolation (Sec. 3.2): Take the $i_{th}$ category as an example, we form a prompt: "a photo of a $[V_1^i]\,[V_2^i]\,...\,[V_n^i]$ [metaclass][2]". The "[metaclass]" is the theme of the corresponding dataset, *e.g.*"bird" is the "[metaclass]" for dataset CUB (Wah et al., 2011). Then, we calculate the DDIM inversion $I_j^i$ for each training sample $X_j^i \in \mathcal{O}^i$ conditioned on this prompt. All these inversions (from $\mathcal{O}^i$) made up the **inversion pool** $\mathcal{I}^i = \{I_j^i\}_{j=1}^{N_i}$ (*c.f.*, Sec. 3.2.1). After that, we randomly sample two inversions $(I_a^i, I_b^i)$ from $\mathcal{I}^i$ and conduct **random circle interpolation** on this pair (*c.f.*, Sec. 3.2.2). The interpolation result is denoted as $Z$. We repeat the sampling and interpolation then collect all interpolation results into $\mathcal{Z}^i$.

3) Two-stage Denoising (Sec. 3.3): Given an interpolation $Z \in \mathcal{Z}^i$, we denoise it as the initial noise in two stages. The main difference between the two stages is the prompt used. In the first stage, we use a **suffixed prompt**: "a photo of a $[V_1^i]\,[V_2^i]\,...\,[V_n^i]$ [metaclass] [suffix]". In the second stage, we use a **plain prompt**: "a photo of a $[V_1^i]\,[V_2^i]\,...\,[V_n^i]$ [metaclass]". Repeat two-stage denoising for each $Z \in \mathcal{Z}^i$, then we can get all the synthetic images and collect them into $\mathcal{S}^i$.

### 3.1 CATEGORY CONCEPTS LEARNING

The pre-trained datasets of diffusion models may have a distribution gap with the downstream classification benchmarks. Thus, it is hard to directly use category labels to guide the diffusion model to

generate corresponding faithful images. Learning a more faithful concept[3] for each category as the prompt for downstream generation is quite necessary. To achieve this, we followed the same learning strategy as (Wang et al., 2024). Specifically, there are two learnable parts: 1) *Token embeddings*: For the $i_{th}$ category, we set $n$ learnable concept tokens ($\{[V_j^i]\}_{j=1}^n$) 2) *Low-rank matrixes*: We insert some low-rank matrixes (Hu et al., 2021) into the pretrained diffusion U-Net. These matrixes are shared by all categories.

Based on the above, given $X_j^i \in \mathcal{O}^i$, its prompt is "a photo of a $[V_1^i]$ $[V_2^i]$ ... $[V_n^i]$ [metaclass]". For timestep $t$ in the forward process of diffusion, the noised latent $x_t$ can be calculated as follows:

$$x_t = \sqrt{\bar{\alpha}_t}x_0 + \sqrt{1 - \bar{\alpha}_t}\epsilon, \tag{1}$$

where $x_0$ is the encoded latent of $X_j^i$, $\bar{\alpha}_t$ is a pre-defined parameter and $\epsilon$ is a Gaussian noise. The learning objective is:

$$\min_{\theta} \mathbb{E}_{\epsilon,x,c,t} \left[||\epsilon - \epsilon_\theta(x_t, c, t)||_2^2\right], \tag{2}$$

where $c$ is the encoded prompt, $\epsilon_\theta$ is the predicted noise of the diffusion model.

### 3.2 INVERSION INTERPOLATION

#### 3.2.1 INVERSION POOL

To get a faithful and diverse synthetic set by interpolating image pairs, we propose to conduct interpolation in the DDIM (Song et al., 2020) inversion space. There are two main motivations: 1) The sampling speed of DDIM is competitive due to the sampling of non-consecutive time steps. This can make our inverse process efficient. 2) We found that starting from the DDIM inversion can ensure a relatively high reconstruction result, especially conditioned on the learned concepts from Sec. 3.1.

The DDIM sampling has the following updating equation:

$$x_{t-1} = \sqrt{\bar{\alpha}_{t-1}}(\frac{x_t - \sqrt{1 - \bar{\alpha}_t}\epsilon_\theta(x_t, c, t)}{\sqrt{\bar{\alpha}_t}}) + \sqrt{1 - \bar{\alpha}_{t-1}}\epsilon_\theta(x_t, c, t), \tag{3}$$

where $x_t$ is the latent at timestep $t$ in reverse process. Based on Eq. (3), we can get the DDIM inversion update equation:

$$x_t = \frac{\sqrt{\bar{\alpha}_t}}{(\sqrt{\bar{\alpha}_t} - 1)}(x_{t-1} - \sqrt{1 - \bar{\alpha}_{t-1}}\epsilon_\theta(x_t, c, t)) + \sqrt{1 - \bar{\alpha}_t}\epsilon_\theta(x_t, c, t) \tag{4}$$

Given a training sample $X_j^i \in \mathcal{O}^i$, we first encode it into $x_0$ with a VAE encoder. Then we leverage Eq. (4) to update $x_t$ while $c$ is the text embedding of "a photo of a $[V_1^i]$ $[V_2^i]$ ... $[V_n^i]$ [metaclass]". When $t$ reaches the maximum timestep $T$, the $x_T$ is the final DDIM inversion. After conducting Eq. (4) for each $X_j^i \in \mathcal{O}^i$, we can construct an **inversion pool** $\mathcal{I}^i = \{I_j^i\}_{j=1}^{N_i}$.

#### 3.2.2 RANDOM CIRCLE INTERPOLATION

Since Gaussian noises are received as input during the training process of the diffusion model, we need to ensure the initial noise for the denoising process also resides in a Gaussian distribution. Since each inversion in $\mathcal{I}^i$ is in a Gaussian distribution, the common linear interpolation will lead to a result that is not in Gaussian distribution. Thus, we propose to conduct circle interpolation on the inversion pairs. This operation has a larger interpolation range (which increases the diversity) and can maintain the interpolation result in Gaussian distribution[2]. Thus, it can be the initial noise for the denoising process.

After getting the inversion set $\mathcal{I}^i$, we randomly select two DDIM inversions $I_a, I_b$ (ignored the superscript) as a pair from $\mathcal{I}^i$. For this pair, we conduct the random circle interpolation.

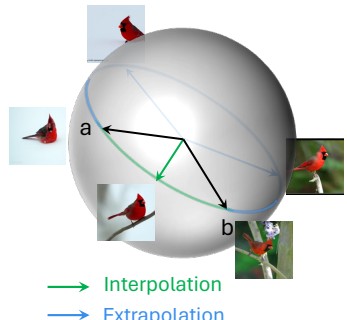

→ Interpolation
→ Extrapolation

Figure 4: Circle Interpolation

---

[3]Typical concept learning techniques like Textual inversion (Gal et al., 2022) or Dreambooth (Ruiz et al., 2023) apply to situations with few samples, which meet the data-scarce sceneries of DA.

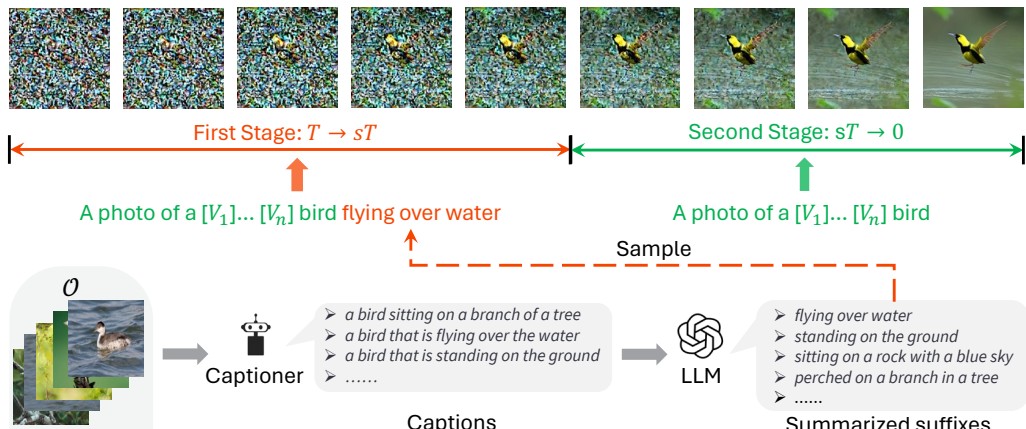

Figure 5: **Two-stage denoising**. Input all images into a captioner and get all captions. Then leverage the language model to summarize these captions into some suffixes. Finally, denoise with the suffixed prompt in the first stage and with the plain prompt in the second stage.

**Circle interpolation.** The circle interpolation can be intuitively understood as rotating from one to another and it can be expressed as follows:

$$Z = \frac{sin((1+\lambda)\alpha)}{sin(\alpha)}I_a - \frac{sin(\lambda\alpha)}{sin(\alpha)}I_b, \qquad \lambda \in [0, 2\pi/\alpha], \tag{5}$$

where $\alpha = arccos(\frac{I_a^T I_b}{(||I_a||||I_b||)})$ and $Z$ is the final interpolation result. $\lambda$ is a random interpolation strength, which can decide the interpolation type (interpolation or extrapolation) and control the relative distance between $Z$ and $I_a$, $I_b$.

As shown in Figure 4, the path of circle interpolation is the circle composed of the Green Arc and the Blue Arc. According to the rotation direction, we can decompose the circle interpolation into spherical interpolation and spherical extrapolation (Shoemake, 1985):

**Spherical Interpolation.** The spherical interpolation means rotate along the shortest path (*c.f.*, the Green Arc of Figure 4) and it can be expressed as follows:

$$Z = \frac{sin((1-\lambda)\alpha)}{sin(\alpha)}I_a + \frac{sin(\lambda\alpha)}{sin(\alpha)}I_b, \qquad \lambda \in [0, 1] \tag{6}$$

**Spherical Extrapolation.** The spherical extrapolation means rotate along the opposite direction of the interpolation path (*c.f.*, the Blue Arc of Figure 4) and it can be expressed as follows:

$$Z = \frac{sin((1+\lambda)\alpha)}{sin(\alpha)}I_a - \frac{sin(\lambda\alpha)}{sin(\alpha)}I_b, \qquad \lambda \in [0, 2\pi/\alpha - 1] \tag{7}$$

According to the periodicity of trigonometric functions, we can see that Eq. (5) is a unified representation of spherical interpolation (Eq. (6)) and spherical extrapolation (Eq. (7)). Based on the expansion rate of the $i_{th}$ category, we repeat the sampling and interpolation. Then, we collect all the interpolation results into $\mathcal{Z}^i$, which will be used as the initial noises in Sec. 3.3.

### 3.3 TWO-STAGE DENOISING

In order to further increase the diversity of synthetic images, we design a two-stage denoising process (*c.f.*, Figure 5). We split the denoising process into two stages with a **split ratio** $s \in [0, 1]$. The first stage includes time steps from $T$ to $sT$. The second stage includes time steps from $sT$ to 0. The main difference between the two stages is the prompt used.

**Suffixed Prompt.** For a specific dataset, we will generate a few suffixes that can summarize the context of this dataset. First, we input each $X_j \in \mathcal{O}$ into a pre-trained vision language model (VLM) (*e.g.*, BLIP (Li et al., 2022)) to get the corresponding caption. After getting all the captions, we input

| Method | 5-shot | | | | | 10-shot | | | | |
|---|---|---|---|---|---|---|---|---|---|---|
| | CUB | Aircraft | Pet | Car | Avg | CUB | Aircraft | Pet | Car | Avg |
| **ResNet50** Original | 54.52 | 36.63 | 86.98 | 40.10 | 54.56 | 70.26 | 54.11 | 90.41 | 69.64 | 71.11 |
| Real-Filter (He et al., 2022) | 55.03 | 30.40 | 87.66 | 57.50 | 57.65(+3.09) | 67.36 | 47.21 | 90.14 | 69.64 | 70.66(−0.45) |
| Real-Guidance (He et al., 2022) | 54.67 | 36.27 | 86.22 | 42.45 | 54.90(+0.34) | 69.81 | 54.77 | 90.06 | 70.97 | 71.40(+0.29) |
| Real-Mix (Wang et al., 2024) | 51.00 | 22.53 | 88.63 | 52.06 | 53.56(−1.00) | 64.65 | 44.16 | 89.40 | 75.72 | 68.48(−2.63) |
| Da-Fusion (Trabucco et al., 2023) | 59.40 | 34.98 | 88.64 | 51.90 | 58.73(+4.17) | 72.05 | 51.14 | 90.47 | 77.61 | 72.82(+1.71) |
| Diff-AUG (Wang et al., 2024) | 61.14 | 39.48 | 89.24 | 62.28 | 63.04(+8.78) | 72.02 | 55.43 | 90.28 | 81.65 | 74.85(+3.74) |
| Diff-Mix (Wang et al., 2024) | 56.18 | 32.61 | 88.77 | 56.39 | 58.49(+3.93) | 70.16 | 52.33 | 90.79 | 78.68 | 72.99(+1.88) |
| **Ours** | **62.22** | **42.15** | **89.84** | **64.24** | **64.61**(+10.05) | **72.66** | **57.43** | **91.01** | **82.02** | **75.78**(+4.67) |
| **ViT-B/16** Original | 73.00 | 32.85 | 90.71 | 59.72 | 64.07 | 83.52 | 51.59 | 92.92 | 81.01 | 77.26 |
| Real-Filter (He et al., 2022) | 73.66 | 31.44 | 90.85 | 72.51 | 67.12(+3.05) | 82.19 | 46.43 | 93.14 | 84.69 | 76.61(−0.65) |
| Real-Guidance (He et al., 2022) | 74.74 | 35.10 | 91.23 | 62.34 | 65.85(+1.78) | 83.24 | 53.15 | 93.22 | 81.97 | 77.90(+0.64) |
| Real-Mix (Wang et al., 2024) | 72.25 | 32.91 | 90.92 | 70.55 | 66.66(+2.59) | 80.95 | 47.01 | 93.16 | 83.78 | 76.23(−1.03) |
| Da-Fusion (Trabucco et al., 2023) | 76.24 | 34.20 | 93.03 | 71.07 | 68.64(+4.57) | 83.97 | 51.68 | 93.68 | 85.00 | 78.58(+1.32) |
| Diff-AUG (Wang et al., 2024) | 77.24 | 40.08 | 92.27 | 76.48 | 71.52(+7.45) | 84.10 | 56.24 | 93.93 | 87.59 | 80.47(+3.21) |
| Diff-Mix (Wang et al., 2024) | 74.49 | 36.48 | 92.29 | 73.01 | 69.07(+5.00) | 81.98 | 53.92 | 93.98 | 85.42 | 78.83(+1.57) |
| **Ours** | **79.22** | **40.23** | **93.79** | **77.10** | **72.59**(+8.52) | **84.89** | **56.32** | **94.25** | **87.81** | **80.82**(+3.56) |

Table 1: **Few-shot classification**. 5-shot and 10-shot results on four fine-grained datasets with two backbones. "Original" means the model trained on the original set without DA. Green and red numbers are increase and decrease values w.r.t. "Original". All results are averaged on three trials.

them into a large language model (LLM) (*e.g.*, GPT-4 (Achiam et al., 2023)) to summarize them into a few descriptions with the following format: "a photo of a [metaclass] [suffix]". Thus, we can get a few suffixes for a dataset[2]. Based on the captioning ability of VLM and the powerful generalization ability of LLM, these suffixes summarize the high-frequency context in the dataset. For each $Z \in \mathcal{Z}^i$, we randomly sample one suffix then concat the plain prompt with this suffix into: "a photo of a $[V_1^i][V_2^i]...[V_n^i]$ [metaclass] [suffix]".

**Denoising Process.** Based on the above, the first stage uses the suffixed prompt while the second stage removes the suffix part. We can express our two-stage denoising process as follows:

$$x_{t-1} = \sqrt{\bar{\alpha_{t-1}}}(\frac{x_t - \sqrt{1 - \bar{\alpha_t}}\epsilon_\theta}{\sqrt{\bar{\alpha_t}}}) + \sqrt{1 - \alpha_{t-1}}\epsilon_\theta \quad \begin{cases} \epsilon_\theta = \epsilon_\theta(x_t, c^*, t), & t \in (sT, T] \\ \epsilon_\theta = \epsilon_\theta(x_t, c, t), & t \in [0, sT) \end{cases}, \quad (8)$$

where $x_T = Z \in \mathcal{Z}^i$, $c^*$ and $c$ are the text embedding of suffixed prompt and the prompt without suffix part respectively. After the above update, we can obtain $x_0$ and then form the synthetic set $\mathcal{S}$.

## 4 EXPERIMENTS

### 4.1 FEW-SHOT CLASSIFICATION

**Settings.** To evaluate the Diff-II's augmentation capacity based on few samples, we conducted few-shot classification on four domain-specific fine-grained datasets: **Caltech-UCSD Birds** (Wah et al., 2011), **FGVC-Aircraft** (Maji et al., 2013), **Stanford Cars** (Krause et al., 2013) and **Oxford Pet** (Parkhi et al., 2012), with shot numbers of 5, 10. We used the augmented datasets to fine-tune two backbones: 224×224-resolution ResNet50 (He et al., 2016) pre-trained on ImageNet1K (Deng et al., 2009) and 384×384 ViT-B/16 (Dosovitskiy et al., 2020) pre-trained on ImageNet21K. We compared our method with six diffusion-based augmentation methods: **Real-Filter**, **Real-Guidance** (He et al., 2022), **Da-Fusion** (Trabucco et al., 2023), **Real-Mix**, **Diff-AUG** and **Diff-Mix** (Wang et al., 2024). We fixed $s$ to 0.3 for 5-shot and 0.1 for 10-shot. For fairness, the expansion rate was 5 for all methods. For the classifier training process, we followed the joint training strategy of (Trabucco et al., 2023): replacing the data from the original set with synthetic data in a replacement probability during training. We fixed the replacement probability with 0.5 for all methods. More details are in the Appendix A.3.

**Results.** From the results in Table 1, we have several observations: 1) Compared with training on the original set, our method can improve the average accuracy from 3.56% to 10.05%, indicating that our methods can effectively augment domain-specific fine-grained datasets. 2) Our method can outperform all the comparison methods in all settings, demonstrating the effectiveness of our method for few-shot scenarios. 3) Our method achieves greater gains for smaller shots (*i.e.*, 5shot) and weaker backbone (*i.e.*, ResNet50), showing our method is robust to challenging settings.

| Method | CUB-LT | | | | Flower-LT | | | |
|---|---|---|---|---|---|---|---|---|
| | IF=100 | IF=20 | IF=10 | Avg | IF=100 | IF=20 | IF=10 | Avg |
| CE | 33.65 | 44.82 | 58.13 | 45.53 | 80.43 | 90.87 | 95.07 | 88.79 |
| CMO (Park et al., 2022) | 32.94 | 44.08 | 57.62 | 44.88(−0.65) | 83.95 | 91.43 | 95.19 | 90.19(+1.40) |
| CMO-DRW (Cao et al., 2019) | 32.57 | 46.43 | 59.25 | 46.08(+0.55) | 84.07 | 92.06 | 95.92 | 90.68(+1.89) |
| Real-Gen (Wang et al., 2024) | 45.86 | 53.43 | 61.42 | 53.57(+8.04) | 83.56 | 91.84 | 95.22 | 90.21(+1.42) |
| Real-Mix (Wang et al., 2024) | 47.75 | 55.67 | 62.27 | 55.23(+9.70) | 85.19 | 92.96 | 96.04 | 91.40(+2.61) |
| Diff-Mix (Wang et al., 2024) | 50.35 | 58.19 | 64.48 | 57.67(+12.14) | 89.46 | 93.86 | 96.63 | 93.32(+4.53) |
| **Ours** | **51.21** | **62.31** | **70.28** | **61.27**(+15.74) | **89.54** | **94.39** | **97.35** | **93.76**(+4.97) |

Table 2: **Long-tail classification results on CUB-LT and Flower-LT**. "CE" is a plain baseline that trains a classifier on the original set with the Cross-Entropy loss. It contains no operations designed for long-tail tasks. "IF" is the imbalanced factor, where a larger IF indicates more imbalanced data distribution. Green and red numbers are the increase and decrease values w.r.t. CE. "Ours" results are averaged on three trials, and other results are from (Wang et al., 2024)

## 4.2 LONG-TAIL CLASSIFICATION

**Settings.** To evaluate the Diff-II's augmentation capacity for datasets with imbalanced samples, we experimented with our methods on the long-tail classification task. Following the previous settings (Cao et al., 2019; Liu et al., 2019; Park et al., 2022; Wang et al., 2024), we evaluated our method on two domain-specific long-tail datasets: **CUB-LT** (Samuel et al., 2021) and **Flower-LT** (Wang et al., 2024), with imbalance factor (IF) of 100, 20,10. We used the 224×224-resolution ResNet50 (mentioned in Sec. 4.1) as the backbone. We fixed $s$ to 1.0 for all settings. We compared our method with five methods: **oversampling-based CMO** (Park et al., 2022), **re-weighting CMO** (CMO+DRW (Cao et al., 2019)), diffusion-based **Real-Filter**, **Real-Guidance**, **Real-Mix**, and **Diff-Mix**. For fairness, the expansion rate was 5, and the replacement probability was 0.5 for all diffusion-based methods. More details are in the Appendix A.3.

**Results.** From the results in Table 2, we have several observations: 1) Our method can outperform all the comparison methods in all settings. For example, the average accuracy on CUB-LT exceeds the previous state-of-the-art Diff-Mix 3.6%, demonstrating our method can well mitigate the imbalanced data distribution. 2) Compared with the case of relatively low imbalanced factors (*e.g.*, IF=10), the gain brought by our method will be reduced when the imbalanced factor is quite high (*e.g.*, IF=100). This is because when the imbalance is too high, there is only one sample for many categories, making our inversion interpolation can not be implemented.

## 4.3 OUT-OF-DISTRIBUTION CLASSIFICATION

**Settings.** To evaluate whether the synthetic data generated by Diff-II can benefit the generalization capacity of the classifier, we conducted out-of-distribution (OOD) classification experiments. To be specific, we trained a 224×224-resolution ResNet50 (*c.f.*, Sec. 4.1) with the original set of 5shot CUB and corresponding synthetic data (same with Sec. 4.1). Then we tested it on an out-of-distribution dataset: **Waterbird** (Sagawa et al., 2019). The Waterbird is constructed in this way: segment the CUB's foregrounds and paste them into the images from Places (Zhou et al., 2017). The images from Places provide new backgrounds for CUB's foregrounds. The Waterbird dataset can be divided into 4 groups: (land, land), (water, water), (land, water) and (water, land). The former in brackets refers to the type of foreground bird (water bird or land bird), and the latter refers to the type of background bird (water background or land background). For example, the (land, water) indicates the land bird in the water background. Besides, the comparison methods were six diffusion-based data augmentation methods: **Real-Filter**, **Real-Guidance**, **Da-Fusion**, **Real-Mix**, **Diff-AUG** and **Diff-Mix**. We used the same hyper-parameters with Sec. 4.1.

**Results.** As shown in Table 3, we can have two observations: 1) Our method can significantly improve the classification ability of the classifier on the background-shift out-of-distribution dataset by augmenting the original dataset. For example, the average accuracy can be improved by 11.39% compared to the "Original" (no augmentation) one. This shows that the data generated by our Diff-II has good diversity, so it is possible to train a classifier that is robust to the background. 2) Our method can outperform all the comparison methods in 4 groups. Especially in (water, land) group, Diff-II can outperform the second-best method (Diff-AUG) by 3.45%. This demonstrates the excellent ability of our method to generate faithful and diverse images.

| Method | land,land | water,water | land,water | water,land | Avg |
|--------|-----------|-------------|------------|------------|-----|
| Original | 38.55 | 32.15 | 37.90 | 30.20 | 34.70 |
| Real-Filter | 38.87 | 34.72 | 38.25 | 31.64 | 35.87(+1.17) |
| Real-Guidance | 39.49 | 33.64 | 39.80 | 29.78 | 35.68(+0.98) |
| Real-Mix | 30.29 | 29.44 | 30.02 | 26.95 | 29.18(−5.52) |
| Da-Fusion | 44.50 | 39.04 | 44.46 | 33.49 | 40.37(+5.67) |
| Diff-AUG | 49.02 | 40.59 | 48.71 | 38.58 | 44.23(+9.53) |
| Diff-Mix | 33.84 | 30.53 | 34.72 | 28.19 | 31.82(−2.88) |
| **Ours** | **49.84** | **41.46** | **51.01** | **42.03** | **46.09**(+11.39) |

Table 3: **OOD classification.** Results are averaged on 3 trials.

| I | E | TD | LPIPS (↑) | Acc (↑) |
|---|---|----|-----------|---------|
| ✓ | | | 35.4% | +1.07 |
| ✓ | ✓ | | 51.5% | +3.86 |
| | | ✓ | 50.0% | +4.15 |
| ✓ | ✓ | ✓ | **52.7%** | **+5.52** |

Table 4: **Components Ablation.** "I" is Spherical Interpolation, "E" is Spherical Extrapolation and "TD" is Two-stage Denosing. "Acc" is the increase relative to no DA.

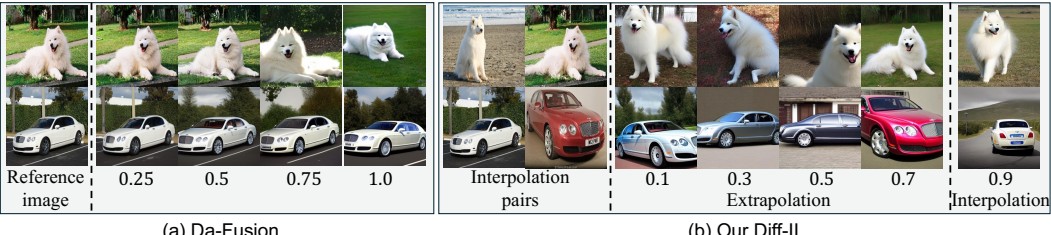

(a) Da-Fusion          (b) Our Diff-II

Figure 7: **Visualization Comparison**. (a) Synthetic images of Da-fusion regarding different translation strengths. (b) Synthetic images of our Diff-II regarding different interpolation strengths (The unit is $2\pi/\alpha$). Experientially, the interpolation type is extrapolation when the strength is in $[0, 0.75]$, else interpolation.

## 4.4    ABLATION STUDY

**Effectiveness of Each Component.** Our Diff-II has two key components: 1) Circle Interpolation, containing the interpolation (I) and extrapolation (E). 2) Two-stage Denoising (TD). We investigated the synthetic set of 5-shot Aircraft (same setting as Sec. 3.1 with ResNet) and reported: average **LPIPS** (Zhang et al., 2018) between images of the synthetic set (which can reflect the diversity), and classification **accuracy**. In the first row of Table 4, independent I can improve the accuracy. By incorporating E in the second row, the diversity and accuracy are further improved because of the larger interpolation space. This indicates that our circle interpolation has a larger interpolation range by combining both I and E. Independently adding TD can also improve the performance. After combining all components, the LPIPS further increased, thus boosting higher accuracy.

**Split Ratio.** Recall that in the Two-stage Denoising (*c.f.*, Sec. 3.3), we have a split ratio $s$ to divide the denoising into two stages. To explore how the split ratio influences the synthetic data, we ablated it in Figure 6. This figure gives the curves of the CLIP score (Hessel et al., 2021) of the synthetic set and average LPIPS between images of the synthetic set changing with $s$. We can see that, with the increasing $s$, the CLIP Score decreases at a relatively slow rate while the LPIPS has a relatively large increase. By adjusting $s$, a trade-off between faithfulness and diversity can be made.

**Qutative Results.** In Figure 7, we give some visualizations of Da-Fusion (Trabucco et al., 2023) and our Diff-II. We can see that the samples generated by DA-fusion lack diversity. In contrast, our Diff-II can generate samples with new context while maintaining the category characteristics.

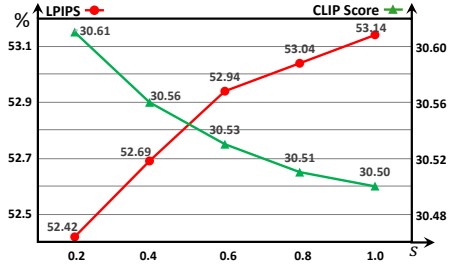

Figure 6: **Influence of split ratio** $s$. Except for the split ratio, all other settings and hyperparameters are the same with 5-shot CUB classification with ResNet50.

## 5    CONCLUSION

In this work, we analyze current diffusion-based DA methods from a unified perspective, finding that they can either only improve the faithfulness of synthetic samples or only improve their diversity. To take both faithfulness and diversity into account, we propose Diff-II, a simple yet effective diffusion-based DA method. our Diff-II show that it significantly improves both the faithfulness and diversity of the synthetic samples, further improving classification models in data-scarce sceneries. In the future, we are going to: 1) extend this work into more general perception tasks, such as object detection, segmentation, or even video-domain tasks. 2) explore more effective DA methods that can better handle situations with extremely few training images.

**Limitations.** Our method is less effective when some categories only have one training image. In that case, the interpolation can not be implemented because the interpolation operation is between two samples. We can see that for the long-tail classification task on CUB-LT (*c.f.*, Table 2): as the imbalance factor gets larger (from 10 to 100), the gain of our method (compared with the second best one Diff-Mix)is getting smaller and smaller (from $5.8\%$ to $0.86\%$). This is because a higher imbalance factor means there are more categories that only have one training image.

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

# A APPENDIX

This appendix is organized as follows:

- Section A.1 gives the basic theory of spherical interpolation and the derived spherical extrapolation. Then we provide proof that the circle interpolation result of two DDIM inversions is approximately standard normal distribution.
- Section A.2 provides details of the comparison methods (*c.f.*, Section 4), including Real-Filter (He et al., 2022), Real-Guidance (He et al., 2022), Da-Fusion (Trabucco et al., 2023), Real-Mix (Wang et al., 2024), Diff-AUG (Wang et al., 2024), Diff-Mix (Wang et al., 2024), CMO (Park et al., 2022) and CMO+DRW (Cao et al., 2019).
- Section A.3 gives the implementation details of our Diff-II and the reproduction details of the comparison methods.
- Section A.4 shows additional results of experiments. First, we give the suffixes and predefined metaclasses of each dataset in Section A.4.1 and Section A.4.2 respectively. Then, we give the more ablation results of the key components and *split ratio* (*c.f.*, Section 3.3) in Section A.4.3. Finally, we give additional visualizations in Section A.4.4.

## A.1 THEORY OF SPHERICAL INTERPOLATION

Spherical interpolation is a method used to interpolate between two points on a sphere or in a spherical space. The main idea behind spherical interpolation is to find a point along the shortest path on the sphere's surface between two given points. The theory of spherical interpolation is grounded in spherical geometry and quaternion algebra as follows:

**Shortest Path on Sphere**: The shortest path between two points on the surface of a sphere is along the great circle that passes through both points. A great circle is any circle that divides the sphere into two equal hemispheres, like the equator or the meridians on a globe.

**Interpolation Formula**: Given two points on a sphere, represented by unit vectors $A$ and $B$, and an interpolation parameter $t \in [0, 1]$, spherical interpolation calculates a third point $P$ along the great circle from $A$ to $B$ using the formula:

$$P = \frac{\sin((1-t)\theta)A + \sin(t\theta)B}{\sin(\theta)}, \tag{9}$$

where $\theta$ is the angle between $A$ and $B$, found using the dot product $\cos(\theta) = A \cdot B$.

**Quaternion Interpolation**: When dealing with rotations in computer graphics, spherical interpolation can be applied using quaternions. Quaternions provide a way to represent orientations and rotations in three dimensions without the singularity and ambiguity problems of Euler angles. The interpolation of two quaternions $q_1$ and $q_2$ is given by:

$$q = \frac{\sin((1-t)\theta)q_1 + \sin(t\theta)q_2}{\sin(\theta)}, \tag{10}$$

where $\theta$ is the angle between the quaternions, computed as $\cos(\theta) = \mathrm{Re}(q_1^* q_2)$ (with $q_1^*$ being the conjugate of $q_1$). Spherical interpolation can smoothly interpolate rotations and directions, ensuring that the interpolated values remain on the sphere, and thus maintaining the integrity of the rotations or directional data.

Based on the above, we can easily derive the spherical interpolation $Z$ between two vectors ($I_a$ and $I_b$) of the same length:

$$Z = \frac{sin((1-\lambda)\alpha)}{sin(\alpha)}I_a + \frac{sin(\alpha\lambda)}{sin(\alpha)}I_b, \qquad \lambda \in [0, 1], \tag{11}$$

where $\lambda$ is the interpolation strength, $\alpha = arccos(\frac{I_a^T I_b}{(||I_a||||I_b||)})$ and $Z$ is the final interpolation result. Then we generalize to spherical extrapolation:

$$Z = \frac{sin((1+\lambda)\alpha)}{sin(\alpha)}I_a - \frac{sin(\alpha\lambda)}{sin(\alpha)}I_b, \qquad \lambda \in [0, 2\pi/\alpha - 1]. \tag{12}$$

Spherical extrapolation can expand the trajectory along the interpolation trajectory, increasing the interpolation range while still maintaining the integrity. Based on the periodicity of trigonometric functions, we can merge spherical interpolation and extrapolation into circle interpolation:

$$Z = \frac{sin((1+\lambda)\alpha)}{sin(\alpha)}I_a - \frac{sin(\alpha\lambda)}{sin(\alpha)}I_b, \qquad \lambda \in [0, 2\pi/\alpha]. \tag{13}$$

Then we give the proof that the circle interpolation of two DDIM inversions is approximately standard normal distribution.

First, we consider that $I_a$ and $I_b$ are two DDIM inversions, which are approximately in standard normal distribution:

$$I_a \sim N(\mu_a, \sigma_a^2), \qquad \mu_a \simeq 0, \sigma_a \simeq 1. \tag{14}$$
$$I_b \sim N(\mu_b, \sigma_b^2), \qquad \mu_b \simeq 0, \sigma_b \simeq 1. \tag{15}$$

According to the superposition of normal distribution, we can get the distribution of $Z$:

$$Z \sim N(\frac{sin((1+\lambda)\alpha)}{sin(\alpha)}\mu_a - \frac{sin(\alpha\lambda)}{sin(\alpha)}\mu_b, (\frac{sin((1+\lambda)\alpha)}{sin(\alpha)})^2\sigma_a^2 + (\frac{sin(\alpha\lambda)}{sin(\alpha)})^2\sigma_b^2). \tag{16}$$

For the mean term of Eq. (16):

$$\frac{sin((1+\lambda)\alpha)}{sin(\alpha)}\mu_a - \frac{sin(\alpha\lambda)}{sin(\alpha)}\mu_b \simeq 0. \tag{17}$$

For the variance term, the $\alpha \simeq \pi/2$ due to $I_a$ and $I_b$ are two independent high-dimension vectors. Thus, $sin\alpha \simeq 1$ and $cos\alpha \simeq 0$. Then, we can simplify the variance term:

$$\begin{aligned}
&(\frac{sin((1+\lambda)\alpha)}{sin(\alpha)})^2\sigma_a^2 + (\frac{sin(\alpha\lambda)}{sin(\alpha)})^2\sigma_b^2 \\
&\simeq sin^2((1+\lambda)\alpha) + sin^2(\alpha\lambda) \\
&= sin^2(\alpha + \alpha\lambda) + sin^2(\alpha\lambda) \\
&= (sin(\alpha)cos(\alpha\lambda) + cos(\alpha)sin(\alpha\lambda))^2 + (sin(\alpha\lambda))^2 \\
&\simeq cos^2(\alpha\lambda) + sin^2(\alpha\lambda) = 1.
\end{aligned} \tag{18}$$

Thus,

$$Z \sim N(\mu, \sigma^2), \qquad \mu \simeq 0, \sigma \simeq 1. \tag{19}$$

Proof completed.

## A.2 COMPARISON METHODS

In this section, we introduce all the comparison methods of experiments.

**Diffusion-based data augmentation methods**:

- *Real-Filter* (He et al., 2022): Directly generate some synthetic images with prompts containing their corresponding category labels. Then, leverage a pre-trained perception network to extract the features of both images of the original training set and synthetic images. Finally, filter all the synthetic images that are far from images of the original training set and only maintain those that are closed to the original training images.
- *Real-Guidance* (He et al., 2022): Given an image from the original training set, add $T$ timesteps noise to the image and use the noised one to replace the random noise at the beginning of the generation. Finally, denoise it with a prompt containing its category label.
- *Da-Fusion* (Trabucco et al., 2023): Firstly, set a few learnable token embeddings to learn an accurate concept for each category with the original training set. Then for a given image of the original training set, add random timesteps noise and denoise the noised image with a prompt containing its learned category concept.
- *Real-Mix* (Wang et al., 2024): Given an image from the original training set, add random timesteps noise to the image. Then denoise the noised image with a prompt containing other-category labels. This will lead to a synthetic image with intermediate semantics between the two categories. Design a calculation mechanism to decide the soft label for this synthetic image.

- *Diff-AUG* (Wang et al., 2024): Firstly, set a few learnable token embeddings and insert some learnable low-rank matrixes into the U-Net to learn an accurate concept for each category with the original training set. Then for a given image, add $T$ timesteps noise and denoise with a prompt containing its learned category concept.
- *Diff-Mix* (Wang et al., 2024): Firstly, set a few learnable token embeddings and insert some learnable low-rank matrixes into the U-Net to learn an accurate concept for each category with the original training set. Then for a given image, add random timesteps noise and denoise with a prompt containing learned other-category concepts. This will lead to a synthetic image with intermediate semantics between the two categories. Design a calculation mechanism to decide the soft label for this synthetic image.

**Long-tail classification methods**:

- *CMO* (Park et al., 2022): To balance the number of different categories's training samples. CMO crops the objects from the rare-category images and pastes them to rich-category images to get some new images with rare-category objects and rich-category images' backgrounds. These new images will be used to expand the rare-category images.
- *CMO+DRW* (Cao et al., 2019): Except on oversampling-based CMO, DRW gives different weights to the loss of different categories. Specifically, the rare categories get a large loss weight while rich categories get a smaller loss weight.

### A.3 IMPLEMENTATION DETAILS

In this section, we give all the implementation details of our Diff-II and reproduction details of comparison methods.

**Details of our Diff-II**:

- *Category concept learning*: We follow the implementations of (Wang et al., 2024)[4].
- *Inversion interpolation*: We use DDIM inversion (Song et al., 2020) with 25 steps and 1.0 guidance scale (Ho & Salimans, 2022) to calculate the inversion for each image. Then for each category, we randomly sample inversion pairs until the number of inversion pairs reaches five times the number of samples in the original training set. After that, we conduct circle interpolation on these pairs with random strength $\lambda \in [0, 2\pi/\alpha]$ (*c.f.*, Section 3.2.2).
- *Two-stage denosing*: We used BLIP-caption (Li et al., 2022) to get captions of all images. Then, we used GPT-4-turbo (Achiam et al., 2023) to summarize the captions into suffixes with the prompt:

  "I have a set of image captions that I want to summarize into objective descriptions that describe the scenes, actions, camera pose, zoom, and other image qualities present.

  My captions are: {captions}

  I want the output to be a $<= 10$ of captions that describe a unique setting, of the form {prefix}.

  Here are 3 examples of what I want the output to look like:

  - {prefix} standing on a branch.
  - {prefix} flying in the sky with the Austin skyline in the background.
  - {prefix} playing in a river at night.

  Based on the above captions, the output should be:"

  Then, for each denoising, we randomly sampled a suffix for the first stage. For 5-shot classification, the split ratio was 0.3; for the 10-shot classification, the split ratio was 0.1; for the long-tail classification, the split ratio was 1.0. For the sample, we used the DDIM sampler with 25 steps and 7.5 guidance scale.

**Details of comparison methods**: For few-shot classification, we followed the reproduction implementations (*i.e.*, the timesteps of adding noise) of (Wang et al., 2024). The translation strengths of Real-Guidance, Real-Mix, Da-Fusion, Diff-AUG, and Diff-Mix are 0.1, random one of [0.5, 0.7, 0.9], random one of [0.25, 0.5, 0.75, 1.0], 1.0, and random one of [0.5, 0.7, 0.9]. For long-tail classification, we directly report the results from (Wang et al., 2024).

---

[4]https://github.com/Zhicaiwww/Diff-Mix

**Details of classifier training**: For fairness, we used 0.5 as the replacement probability for all methods. Besides, we followed (Wang et al., 2024) for other settings and hyperparameters.

**Hardware**: All experiments are conducted on 8 NVIDIA GeForce RTX 3090 GPUs.

## A.4 ADDITIONAL RESULTS

### A.4.1 SUFFIXES

The suffixes of each dataset are listed as follows:

**5-shot CUB**:

- standing on a tree branch.
- flying around flowers.
- standing on a post by the water.
- flying over water.
- standing on the ground.
- swimming in the water.
- sitting on a rock with a blue sky.
- perched on a branch in a tree.
- flying over water with wings spread.
- perched on a tree branch.

**10-shot CUB**:

- flying over water.
- standing on the ground.
- sitting on a rock.
- swimming in water.
- sitting on a bird feeder.
- standing on the beach near water.
- perched on a wire with a blue sky in the background.
- standing on a branch with tall grass in the background.
- flying in the sky with its wings spread.

**5-shot Car**:

- parked on a city street.
- on a white background.
- parked in a lot with green trees in the background.
- parked on a gravel road with mountains in the background.
- driving down a tree-lined road.
- parked on a black floor.
- with its doors open.
- charging at a station.
- driving on a racing track.

**10-shot Car**:

- parked on a road.
- parked on a gravel road.
- parked with trees in the background.
- driving down a forested road.
- driving on a dirt road in desert area.
- on display at a show.
- driving on a city street.
- parked in a garage.
- parked in front of a store with other cars.

**5-shot Aircraft**:

- parked on the runway.
- flying in the sky with the landing gear down.
- landing with another plane in the background.
- on the runway at an airport.
- on the tarmac with mountains in the background.
- flying in the air with the landing gear down.
- parked in a hangar with the door open.
- flying in the sky with palm trees in the background.
- flying in the sky against a blue background.
- lined up on the runway at the airport.

**10-shot Aircraft**:

- flying in the sky with landing gear down.
- taking off from the airport with a city in the background.
- at the tarmac of an airport with a building in the background.
- with passengers, flying in the sky.
- propeller plane on a runway, with a Honeywell sign in the background.
- airplane where workers are seen working on it in a hangar.
- plane with a green stripe on the runway.
- with people on board, floating in water.
- with a red cross on its tail and landing gear.
- jet on the runway with smoke coming out of it.

**5-shot Pet**:

- lying on a pillow on the floor.
- playing with a toy on the floor.
- in the grass, looking at the camera with a leash.
- on a window sill looking out.
- sitting on a couch with a stuffed animal.
- on a rock beside a person.
- running in the grass with a frisbee.

**10-shot Pet**:

- laying down in the grass.
- sitting on brown leather furniture.
- sitting on a couch with a dark background.
- playing with a ball in the grass.
- sitting on a windowsill, looking outdoors.
- standing on a wooden deck.
- sitting inside a cage.
- sitting on a chair with its mouth open.
- laying on a couch with a white background.

**CUB-LT/IF=10**:

- flying over the ocean.
- sitting on a rock by the water.
- standing in the grass.
- flying in the sky with its wings spread.
- swimming in the water.
- standing on a sandy beach.
- sitting on a wire fence.
- perched on a bird feeder in the snow.
- standing on a tree stump.

**CUB-LT/IF=20**:

- sitting on a rock in the water.

- perched on a branch in a tree.
- standing on the ground in the grass.
- sitting on a post by the water.
- standing on a ledge near the water.
- flying in the sky with its wings spread.
- sitting on a branch with a blurred background.

**CUB-LT/IF=100**:

- flying in the sky with its wings spread.
- standing on the ground in the dirt.
- sitting on a branch of a tree.
- swimming in the water.
- perched on a hand in a grassy field.
- standing on the shore of the water.
- sitting on a branch of a tree.
- sitting on a ledge by water.
- standing in the water with its reflection.

**Flower-LT/IF=10**:

- with water droplets on it.
- growing in a garden.
- in a close-up view.
- with a bee on it.
- in front of a water body.
- against a brick wall.
- with a butterfly on it.
- with mixed colors in a bush.

**Flower-LT/IF=20**:

- close-up with a dark background.
- blooming in a garden.
- growing in a pot.
- floating on water in a pond.
- arranged on a table.
- with a bee on it in the garden.
- on a tree.
- in a field with a rocky surface.
- against a blue sky.
- with a blurry background in a field.

**Flower-LT/IF=100**:

- with water droplets on it.
- growing in a garden.
- in a close-up view.
- with a bee on it.
- in front of a water body.
- against a brick wall.
- with a butterfly on it.
- with mixed colors in a bush.

### A.4.2 PREFDEFINED METACLASSES

We list the metaclass of each dataset here: CUB→"bird"; FGVC-Aircraft→"aircraft"; Stanford-Cars→"car"; Oxford-Pet→"animal"; CUB-LT→"bird"; Flower-LT→"flower".

### A.4.3 MORE ABLATION RESULTS FOR SPLIT RATIO

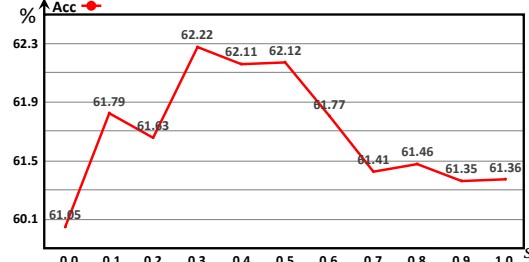

Figure 9: Synthetic images regarding different interpolation strengths (The unit is $2\pi/\alpha$).

**Components**: We ablated our key components: Inversion Interpolation (II) and Two-stage Denoising (TD). We investigated the synthetic set of 5-shot Aircraft (same setting as Sec. 3.1 with ResNet) and reported: **CLIP Score** (Hessel et al., 2021) of the synthetic set; average **LPIPS** (Zhang et al., 2018) between images of the synthetic set, and classification **accuracy**. The CLIP score can reflect the faithfulness of the synthetic set while the LPIPS can indicate the diversity. As shown in Table 4, the first row (w/o both II and TD) directly denoise a random noise with a prompt without suffix in one stage(*c.f.*, Sec. 3.3). We can see that: independently adding II or adding TD both can increase

| II | TD | CLIP Score (↑) | LPIPS (↑) | Acc (↑) |
|----|----|---------------|-----------|---------|
|    |    | **30.73** | 47.9% | +2.11 |
| ✓  |    | 30.65 | 51.5% | +3.86 |
|    | ✓  | 30.63 | 50.0% | +4.15 |
| ✓  | ✓  | 30.60 | **52.7%** | **+5.52** |

Table 5: **Components Ablation.** "II" is Inversion Interpolation and "TD" is Two-stage Denosing. "Acc" is the increase relative to no DA.

the LPIPS while nearly maintaining the CLIP Score. After adding both components together, the LPIPS further increased. This indicates that each component can significantly benefit the diversity with negligible harm to faithfulness, thus boosting higher accuracies. Then we provide some explanations why starting with an interpolation result (the second row in Table 5) is better than a random noise (the first row in Table 5): Interpolation can not only sample some points in latent space that are not easy to sampled by standard normal distribution, but also the relative distance between these points will not be too close. This ensures the improvement of diversity. Besides, according to the characteristics of circle interpolation, these points are in the position with relatively dense semantics of the pre-trained diffusion model, thus ensuring faithfulness. Therefore, the inversion interpolation results tend to generate more diverse samples than random Gaussian noise and can finally bootstrap better classification results.

**Split ratio** We ablated the *split ratio* $s \in [0, 1]$ in Figure 8. we can see that: the value of $s$ will influence final classification accuracy. We get the best balance (when $s = 0.3$) between faithfulness and diversity.

### A.4.4 Additional Visualizations

**Visualizations across different interpolation strength** $\lambda$: As shown in Figure 9, we give our synthetic images regarding different interpolation strengths (*c.f.*, Sec. 3.2.2). We can see that our Diff-II can generate samples with new context while maintaining the category concept characteristics. The interpolation strengths $\lambda$ can control the relative similarity between the synthetic sample and two samples of interpolation pair.

Figure 8: Classification accuracy for different split ratios. Except for the split ratio, all other settings and hyperparameters are the same with 5-shot CUB classification with ResNet50.

**Synthetic images in few-shot classification and long-tail classification**: we gave more synthetic images of our Diff-II used in few-shot and long-tail classification (*c.f.*, Figure 10).

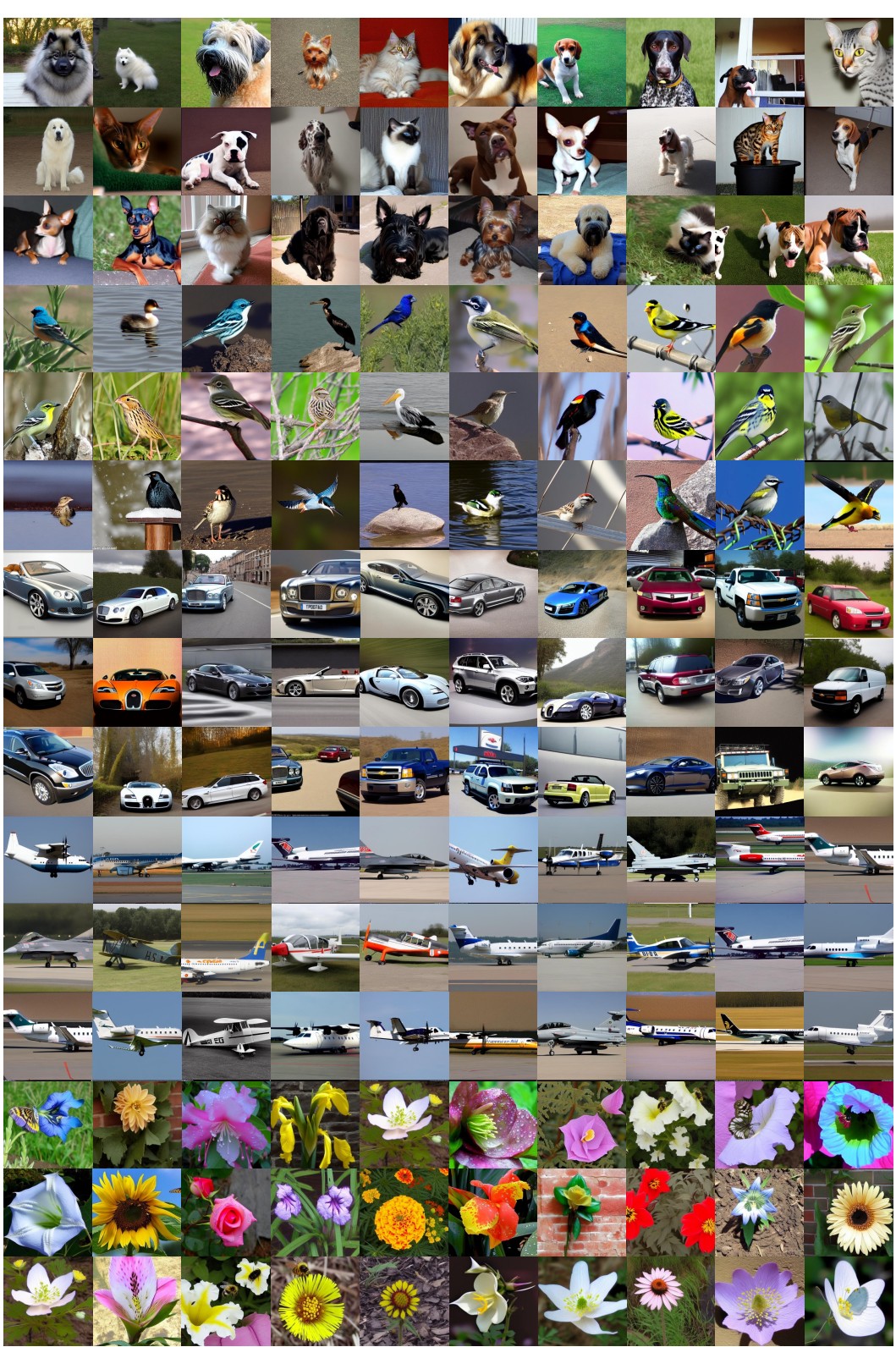

Figure 10: More synthetic images of our Diff-II in few-shot and long-tail classification.

