# OpenReview forum: "Improving Diffusion-based Data Augmentation with Inversion Circle Interpolation"
_ICLR.cc/2025/Conference — ICLR 2025 Conference Withdrawn Submission_

### Official Review · Reviewer_FTug · 2024-11-02

**Soundness:** 3
**Presentation:** 3
**Contribution:** 2
**Rating:** 6
**Confidence:** 3

**Summary:**

This paper proposes a diffusion-based inversion interpolation method for data augmentation, called Diff-II. It first learns a set of learnable token embeddings as concepts for each category. Then, using the learned concepts in the previous step, DDIM inversion is performed for each image to get the latent in the DDIM inversion space. Circle interpolation is adopted to get the latents for new samples using two latents from existing samples in the category. Finally, a two-stage denoising is applied to get the synthetic images in order to reach faithfulness and diversity. Experimental results show that the proposed method outperforms other diffusion-based data augmentation methods on multiple few-shot classification benchmarks.

**Strengths:**

++ The proposed method is simple but effective, giving a decent solution to generate faithful yet diverse samples from diffusion models as data augmentation for improving few-shot classification.

++ Experimental results show that the proposed methods outperform current state-of-the-art diffusion-based solutions like DA-Fusion and Diff-Mix, further pushing forward this research field.

++ The paper presents the theory for spherical interpolation in the appendix, which strengthens the soundness of the proposed method.

++ The presentation of the method is clear and easy to understand.

**Weaknesses:**

-- The paper might overclaim their contribution in "analyzing" current diffusion-based methods to find their synthetic data to either lacking faithfulness or lacking diversity. In fact, this paper only explains in text from Lines 84-98 claiming the current work DA-Fusion has limited diversity, and Diff-Mix lacks faithfulness, and a few visualization comparisons in Figure 7. However, there are no specific experiments demonstrating that the shortcomings of these methods are from faithfulness or diversity of the generated data. Therefore, I think that the conclusion of the issues in current diffusion-based data augmentation methods can only be called "assumptions" or "claims", instead of coming from "analysis".

-- The suffixes in the two-stage denoising are summarized from VLMs to capture the high-frequency context in the dataset. Although this might work for the validation/test set in these specific datasets, it may not work well if we want to build a robust and reliable system handling in-the-wild data in real world application scenarios. Therefore, I think the application of this method might be limited.

-- The paper only compares with diffusion-based augmentation methods, how about non-diffusion-based data augmentation methods, like [1]? Are the diffusion-based methods already the current state-of-the-arts for data augmentation?

[1] Zhou et al. Training on Thin Air: Improve Image Classification with Generated Data. arXiv:2305.15316.

**Questions:**

-- For the purpose of generalization, I think we may not even need VLMs to summarize the context in the dataset. In principle, pretrained diffusion models should have enough knowledge to generate diverse data that can be elicited by diverse text prompts. How is the performance when, for example, using LLMs to generate more diverse prompts instead of using VLMs to summarize from training dataset? I think this should be a more robust solution for building a reliable system in real world applications, although it might have sub-optimal performance on the validation data in the datasets, because the validation/test data in these datasets can also be regarded as in-distribution data.

-- In the experiments when the imbalance factor gets larger (e.g., IF = 100), the paper mentioned in Lines 498-501 that the reason for the performance gain gets smaller is that there are more categories that only have one image, so that there is no way to interpolate between two images. In these cases, do the authors not perform augmentation at all, or the authors have performed augmentation by, for instance, using some DDIM inversion on the single image with different text prompts in the denoising steps?

-- Typos: Line 481: "Qutative" -> "Qualitative". Line 491: "our Diff-II show" -> "Our Diff-II shows".

---

### Official Review · Reviewer_BhjX · 2024-11-03

**Soundness:** 3
**Presentation:** 3
**Contribution:** 2
**Rating:** 6
**Confidence:** 4

**Summary:**

This paper proposes a novel data augmentation approach, Diff-II, to address limitations in existing diffusion-based augmentation methods, which often struggle to balance faithfulness and diversity in synthetic samples. Diff-II introduces three main steps: category concept learning, inversion interpolation using a random circle interpolation, and a two-stage denoising process. The method generates high-quality augmented images with both category-specific details and diverse contexts, enhancing the generalization ability of downstream classifiers. Empirical results on few-shot, long-tail, and out-of-distribution classification tasks demonstrate that Diff-II outperforms existing diffusion-based augmentation methods across multiple benchmarks.

**Strengths:**

- The paper includes a comprehensive set of experiments on various tasks and datasets, with clear performance improvements over state-of-the-art methods.

- The methodology and key concepts are clearly explained, and the visual illustrations support understanding of the augmentation process.

- By generating diverse and category-faithful samples, Diff-II addresses a critical challenge in data augmentation, making it valuable for low-data and imbalanced scenarios.

**Weaknesses:**

- While the circle interpolation provides diversity, it may struggle to capture subtle, fine-grained details in highly complex categories. This potentially leads to synthetic images that lack the nuanced features necessary for representing intricate or visually rich categories, affecting faithfulness.

- The effectiveness of Diff-II relies on accurately learned category embeddings. If these embeddings do not sufficiently capture the characteristics of each category—especially in cases of fine-grained distinctions—the generated samples may diverge from the intended category, which could impact classifier performance.

- The method’s two-stage denoising and complex interpolation steps introduce notable computational demands, especially in high-dimensional datasets. This could limit scalability for applications where rapid data augmentation is essential.

- Diff-II's reliance on interpolation between samples requires a minimum number of data points per category. In cases with extremely sparse data (e.g., only one sample per category), the method’s benefits are diminished, as interpolation cannot be effectively applied, limiting its utility in extreme few-shot scenarios.

**Questions:**

- Could you elaborate on the computational requirements for the two-stage denoising process, especially in high-dimensional datasets?

- How does Diff-II handle cases where categories have minimal data, such as only one sample?

- For datasets with highly variable contexts within a category, does the method maintain both diversity and faithfulness effectively?

- Are there any planned extensions of Diff-II to tasks beyond classification, such as object detection or segmentation?

---

### Official Review · Reviewer_f9DF · 2024-11-04

**Soundness:** 2
**Presentation:** 3
**Contribution:** 1
**Rating:** 3
**Confidence:** 4

**Summary:**

This paper proposes a new diffusion-based data augmentation method, Diff-II, which enhances both the faithfulness and diversity of generated samples. Diff-II comprises three main steps: Category Concepts Learning, Inversion Interpolation, and Two-Stage Denoising. Category Concepts Learning ensures the faithfulness of generated images, while Inversion Interpolation and Two-Stage Denoising improve their diversity.

**Strengths:**

1. The paper is well-organized, making it easy to understand and follow.
2. The problem addressed in this work is valuable and significant.

**Weaknesses:**

1. The pipeline of the proposed data augmentation method is very complex, involving a pre-trained text-to-image diffusion model, a caption model, and a large language model. And the paper reads more like a technical report on how to use these models.
2. The paper lacks sufficient ablation studies to elucidate the effectiveness of each component.
3. The experiments are inadequate to demonstrate the proposed method's effectiveness. Although the primary contribution of this work is a data augmentation method, it does not compare its performance with traditional data augmentation methods on image classification tasks (e.g., ImageNet).

**Questions:**

1. Eq (4) appears to be incorrect.
2. Could you please provide a detailed explanation of DDIM inversion?
3. How do you ensure that the interpolation results between the two DDIM inversions, I_a and I_b, still belong to the same category? Does the text embedding play the primary role in this process?

---

### Official Review · Reviewer_h1M7 · 2024-11-05

**Soundness:** 1
**Presentation:** 2
**Contribution:** 1
**Rating:** 3
**Confidence:** 5

**Summary:**

This paper presents an approach for stable-diffusion-based data augmentation through category concepts learning, inversion interpolation, and multi-stage denoising. By using the pipeline, stable diffusion models can generate more faithful and diverse images which can be used for downstream tasks.

**Strengths:**

1. The paper is easy to follow and understand.
2. Experiments show decent improvement on few-shot fine-grained datasets.

**Weaknesses:**

1. The evaluation scenario seems to be unrealistic, i.e, no evaluation on large-scale datasets. For generative model based data augmentation, existing works [1, 2] already evaluate on ImageNet-scale. Surprisingly, none of this line of work is even mentioned in related work.
2. Existing work like [2] shows with simple prompt diversification, synthetic images can improve both in-domain and out-of-domain at a large scale, which makes me wonder what's the point of doing all these tuning procedure in this work.

3. The long-tail experiment is not promising because the long-tailed classes do not reflect true tail classes in reality. It's just the classes randomly sampled in existing datasets.

[1] Synthetic Data from Diffusion Models Improves ImageNet Classification
[2] Diversify, Don't Fine-Tune: Scaling Up Visual Recognition Training with Synthetic Images

**Questions:**

See above

---

### Note · Authors · 2024-11-14

I have read and agree with the venue's withdrawal policy on behalf of myself and my co-authors.